# Genetic Parameters for a Weighted Analysis of Survivability in Dairy Cattle

**DOI:** 10.3390/ani13071188

**Published:** 2023-03-28

**Authors:** Michaela Černá, Ludmila Zavadilová, Luboš Vostrý, Jiří Bauer, Jiří Šplíchal, Jan Vařeka, Daniela Fulínová, Michaela Brzáková

**Affiliations:** 1Department of Animal Breeding and Genetics, Institute of Animal Science, 104 00 Prague, Czech Republic; 2Plemdat-Genetic Evaluation Department, Czech Moravian Breeders’ Corporation, 252 09 Hradištko, Czech Republic

**Keywords:** survivability, weighted analysis, genetic parameters, Holstein cattle, repeatability model

## Abstract

**Simple Summary:**

In dairy cattle, a cow’s productive life is an essential functional trait with great economic significance and is part of the breeding objectives and comprehensive selection indices. Thus, a highly reliable breeding value prediction is crucial to allow sufficient selection of animals with high genetic potential for survival. Suppose we proceed to evaluate survival in successive periods of an animal’s life. In that case, we encounter an unfavourable phenomenon where the earlier an animal is culled, the more the reliability of its breeding values is underestimated. We assume that this underestimation can be corrected by using a weighted analysis that balances the amount of information regardless of the number of survived periods. This paper attempts to show the methodological procedure for determining the weights and estimating the genetic parameters for the weighted analysis.

**Abstract:**

The genetic parameters for the survival of Holstein cows, analysed in nine consecutive time periods during the first three calving intervals, were estimated. The earlier the animals are culled, the more they are informationally underestimated. This undervaluing can be remedied by using a weighted analysis that balances the amount of information. If the method of estimating breeding values changes, the genetic parameters will also change. The Holstein cattle dataset from 2005 to 2017 used in this study included 1,813,636 survival records from 298,290 cows. The pedigree with three generations of ancestors included 660,476 individuals. Linear repeatability models estimated genetic parameters for overall and functional survivability. Due to weights, heritability increased from 0.013 to 0.057. Repeatability with weights was 0.505. The standard deviations of breeding values were 1.75 and 2.18 without weights and 6.04 and 6.20 with weights. Including weights in the calculation increased the additive variance proportion and the breeding values’ reliabilities. We conclude that the main contribution of the weighted method we have presented is to compensate for the lack of records in culled individuals with a positive impact on the reliability of the breeding value.

## 1. Introduction

In dairy cattle, the length of a cow’s productive life is an essential functional trait with great economic significance [1,2,3] and is part of the breeding objectives and comprehensive selection indices [4,5]. Predicting future survival at an early age can reduce unprofitable investment in an inferior animal [6]. A distinction is made between actual longevity as the number of days lived and functional longevity, i.e., the cow’s ability to delay involuntary culling due to health reasons. Longevity is strongly influenced by the breeder’s voluntary decision to cull an animal based on performance, so dairy cow survival is corrected for milk yield when assessing functional longevity. Survivability indicators used for dairy cows are, e.g., herd life, length of production life (from first calving to culling), and the number of lactations. The survivability indicator is used to assess or estimate the longevity of animals. The evaluation of individuals is intended to reflect how much of the selected period of life they have lived compared with other individuals reared under the same conditions [7]. For example, it expresses the ability to survive from the current lactation period to the next. Survivability is a complex trait, as cows may be discarded from the herd for various reasons [8]. These reasons are low milk production, diseases (mastitis, lameness, metritis, and others), injury, infertility, or death [9,10,11,12]. Survivability has a low heritability with a low but permanent response to selection [13].

For many years, there has been a global trend towards shorter cow lifespans [7,14,15], especially in countries with high milk yields [8,16]. Voluntary culling of cows is the decision of farmers. Cows may be culled during their first lactation period due to health problems and low production, which means an enormous loss of revenue for farmers [16]. Only a minority of cows survive until their fourth lactation period, meaning most are discarded before reaching their maximum potential [17]. Taking functional traits (longevity, health traits, etc.) into account during breeding should lead to an increase in the productive life of cows [16]. Increasing cow longevity would reduce healthcare costs, increase profitability, and likely improve cow welfare and quality of life [18].

Depending on how longevity or survival is expressed, the heritability values differ. For short repetitive periods, which the animal survives over time, the heritability values of individual periods tend to be lower, but with more repeated observations than when considering a long period as a single indicator. For the short repetitive periods that the animal survives, the values of the heritability of individual periods tend to be lower if we compare them with the heritability of the whole period including these short periods. The evaluation of shorter recurrent periods is preferred, as it can better separate the observed data from systematic environmental effects such as herd–year–season (HYS). This is evident for all traits, for example for test-day models (TDM) for milk yield [19], where the heritability value of milk yield for a test-day is much lower than for the whole lactation period, or TDM for animal growth [20]. Van Pelt et al. [13] estimated a low heritability for productive lifespan (0.2% to 3.1%) measured at 1, 3, 6, and 12 months after the first calving. For life expectancy measured 72 months after the first calving, this estimate increased (from 11.5% to 14.9%). Similarly, low heritability was found for heifer survival (1%) and functional longevity in dairy cows (6%) [21]. Samoré et al. [22] estimated the heritability of functional longevity as 6%. Páchová et al. [23] found a heritability of 4.1% in Czech Holstein cows using survival analysis. Zavadilová and Štípková [24] reported heritability for the length of productive life measured in days at 3%, but 5% when measured as the number of lactations initiated. The low estimates of heritability result from low genetic and relatively high residual variability, which can be explained by the complexity of the traits and the sizeable influence of management [25]. However, low genetic variability may be a remnant resulting from different genetic causes of culling, which then falls into residual variability. In general, low heritability coefficients for longevity or survival are also the reason for the low success of selection to increase them.

A standard procedure used for evaluating individuals is the survival analysis implemented in software such as the “Survival kit” [26], which is based on nonlinear hazard functions using Weibull distributions. Other procedures include a linear model with repeatability [27], a linear model with repeatability augmented with random regressions [13,28], and the threshold model [29]. Heise et al. [30] showed that the genetic background of survivability varies across cows and genetic basis of survivability varies also within calving intervals. This leads to multi-trait linear statistical models in which the individual segments of the calving interval and lactation order are treated as genetically distinct traits (nine segments in total). The appropriateness of the individual assessment procedures also depends on whether only the first or subsequent lactations are used or whether overall or functional survivability is assessed. Correlations between breeding values (BVs) according to different prediction methods ranged from 0.7 to 1 [27]. In the Czech Republic, a nationwide genetic assessment of longevity has been conducted using the survival kit method [31]. Incorporating molecular genetic data into an individual’s overall genetic evaluation is essential [32]. In the Czech Republic, the routinely used method for the genetic evaluation of Holstein cattle is the single-step genomic evaluation method (ssGBLUP). Methods for the genetic evaluation of different traits and their use in animal husbandry and breeding conditions are undergoing continuous development [33].

Generally, evaluating the genetic basis for survivability is complex because the actual phenotypic expression of the trait (herd life, production life, etc.) is only discovered later when the animal dies or is retired from breeding. Ways to predict survivability values at a young age are being sought. For example, selection indices include correlated external conformation and other traits [34]. When assessing repeated successive life periods, individuals previously discarded have fewer records in the datasets and, consequently, biased BV and reliabilities. The BV is a ‘regressed’ value, meaning that individuals with low reliability have a BV clustered around the mean, and the variability of the BV increases only with higher reliabilities. For individuals with a small number of records, the genetic evaluation of both themselves and their contemporaries is distorted, and the ranking may change (distortion of genetic breeding values). However, whether there is a following record for an individual depends on the value of the binary traits received in the previous period. Record counts and survivability values are dependent variables, there are ‘non-randomly’ different record counts per individual, and long-lived individuals have different record counts than short-lived individuals. Culled cows receive a final decision and are not assessed further. Individuals who have not been culled are assessed further; their survival records for subsequent life periods increase, and the reliability of its assessment is also gradually increased. The reported reliability of the BV of a discarded individual should be at the same level as that of an individual with survival records for all segments over the entire evaluation period. One option to remove bias in assessing culled individuals could be to assign ‘weights’ to individual observations.

The inclusion of weights substantially increases the volume of information. At the same time, the variability and differences between individuals change. Therefore, new population genetic parameters for overall and functional survivability must be established. The objective of this study was to estimate genetic parameters for the weighted analysis of survivability in dairy cattle using a linear repeatability model and weights assigned to survival periods in Holstein cows.

## 2. Materials and Methods

Following the methodologies proposed by Heise et al. [30], we used nine consecutive periods covering the first three calving intervals, each divided into three periods. An individual goes through these nine periods during their life, which it will or will not survive. Survival is marked ‘100’ and non-survival ‘0’. At each period, the individual encounters contemporaries within the herd–year–season (HYS). Weights were determined on a small artificial set of 10 different individuals. The first individual did not survive the first period and had only one record; only the tenth individual survived all nine periods and had nine records. Each period of an individual’s life was in a different HYS period. The size of all HYSs was complemented to 20 individuals.

Weights (Figure 1 and Table 1) were assigned to survival data in two different ways:

w1: A weight based on the number of non-survived periods. For the all-survived period, a weight of one was given. The record of the non-survived period was weighted by the number of periods in which the animal could no longer survive. An individual culled in the first period was given a weight of nine. An individual culled in the second period was given a weight of one for the first and eight for the second period. An individual culled in the eighth period received a weight of two for that period and 1 for periods 1 to 7. Individuals who survived to the ninth period, whether culled or not, were weighted by one for all periods. The weights for a sample of ten individuals successively culled were 9, 8, 7, 6, 5, 4, 3, 2, 1, and 1.

w2: Weight by the effective number of cases.

The assumption was to achieve by using weights the same reliability for BV of each of the ten individuals no matter how many periods they survived. The reliability of BV depends on the effective number of cases. Here, w depends on the number of contemporaries in each group.
w = 1∗nv/(1 + nv),(1)
where w is the effective number of cases in each HYS effect level, and nv is the number of contemporaries in each HYS effect level.

The sum of w for all known segments of the discarded individual is equal to the sum of the individual with all known periods (∑w). In the period where an individual was culled, the effective number of cases (w_d_) would be the sum of all periods (∑w) minus the effective counts achieved by the individual in the previous periods. The weight of the record is as follows:w = m∗w_d_/(m − w_d_),(2)
where w is the weight of the record at the period in which the individual is culled, m is the number of contemporaries in the given HYS, and w_d_ is the effective number of cases in the culling period.

The integer rounded weights for a sample of ten individuals who were successively culled were 16, 13, 10, 8, 6, 5, 3, 2, 1, and 1.

For the weights calculation: The dataset was analysed using a simple linear model with repeatability, herd, year, season fixed effects (using HYS), and random genetic individual effects (ANIg).
Y = X.HYS + Z.ANIg + e(3)
where Y is the known vector of observed values (the dependent variable);

X is the design matrix of fixed effects;

Z is the design matrix of random effects; 

e is the vector of random errors.

The ten individuals were unrelated. Based on the results published by Heise et al. [30], we modelled heritability = 0.02. The reliability of the BV of the individual (i) is based on the diagonal element of the inverse matrix of the system.
r^2^i = 1 − k∗cii,(4)
where r^2^_i_ is the BV reliability of individual I; 

k is the ratio (1 − h^2^)/h^2^;

c_ii_ is the diagonal element for individual i from the inverse matrix of the system.

Table 1 shows the BV values and their reliabilities for the sample of ten individuals, depending on the weighting method (w1; w2).

Both weighting methods have essentially changed the differences among individuals, better reflecting their biological nature. Individual I (culled in the ninth period) should be genetically equal to individual J, which survived the ninth period. They show good survival with the same positive BV. Individual A did not survive the first period and should be very different from individual B that was culled in the second period and had a more negative BV. The above assumptions were met by weighted evaluation. Individuals I and J survived till period nine, had a weight of ‘1’ in all periods, and achieved an identical BV reliability of r^2^ = 0.15 in all three presented examples. There was also the same difference in BV among the three methods, at approximately 1.73. For other individuals, the BV reliabilities and differences between individuals varied depending on how the weights were used. Weighting changed the overall BV layout and equalised BV reliability. Higher weights (w2) yielded better results, and only w2 will be considered in the rest of the manuscript.

The national population of Holstein cattle up to 2017 was used to determine genetic parameters (5,039,625 records from 903,340 cows in the years 1992–2017). Survivability was monitored for nine consecutive periods during the first three calving intervals, which were divided into three periods according to Heise et al. [30]: the first period up to 49 days after calving, the second period 50–249 days after calving, and the third period above 249 days after calving (the third, sixth, and ninth periods ended by subsequent calving). The third period is of varying length depending on the actual calving period of the cow. The average calving interval was 407 days, with a maximum of 500 days. In the dataset, the records of survival of a given period were marked ‘100’ and of non-survival ‘0’. There were one to nine records per cow, depending on the number of periods it survived.

The dataset size was adjusted to make it possible for the calculation of genetic parameters, which were then estimated. The most recent data were used; cows born after 2005 that first calved up to and including 2014 were used to allow them to survive all nine periods when the data were retrieved in 2014 (we did not use censored data that could bias genetic parameters). HYS was chosen as the calendar quarter within the herd and the year. In HYS, cows from all nine periods were contemporaries of each other. Daughters from sires with fewer than 15 daughters and HYS with fewer than 15 cows were excluded. Editing was performed iteratively until the resulting numbers were stable. During culling, it was checked that each retained cow had an unbroken series of continuous records. The numbers of cases in the adjusted dataset are listed in Table 2.

The dataset included 1,813,636 survivability records from 298,290 cow-daughters of 2329 sires in 15,919 HYS. On average, there were 128.07 daughters per sire, with a range of 15 to 4696. On average, there were 113.93 records per HYS, ranging from 15 to 650. The weights (w2) were assigned to the survival records. On average, the weight was 1.72, and the sum of the weights in the entire dataset for all 1,813,636 records was 3,119,557.

With regards to the proportion of records, 16.45% were in the first period of the total records. The share decreased by period order, with 5.83% of the records in the 9th (last) period. Survivability is an alternative trait with a binomial frequency distribution. The average ‘non-survival’ for all periods was 12.29%, with a standard deviation of 32.8. Survivability was lower in the second and third calving intervals than in the first period. During each calving interval, the culling rate rose sharply, and in the third period, it was several times higher than that in the first. The standard deviations of survivability by period (in parentheses) ranged from 18.2 to 45.2, lowest in the first period of the calving interval and highest in the last period.

For weights (w2), the proportions of the records for each period are different. It should be noted that the higher the weights were, the stronger the influence of the weights on the analysis. The non-survival rates were significantly different. The average survival rate was 49.01%, with a standard deviation of 50.0. The culling rate at particular periods ranged from 14.49% to 69.66%. The differences in the culling rates between the periods decreased. The highest culling rate occurred in the first calving interval, and the lowest was in the third (without weights, the resulting situation was the opposite). The standard deviations of the survival rates by individual periods ranged from 35.2 to 49.9 when weights were considered. A pedigree of three generations of ancestors was linked to a dataset of performance traits that included 660,476 individuals.

### Statistical Model

The analysed traits were survivability and functional survivability [SURV], functional survivability [SURVm], weighted survivability [SURVw2], and functional weighted survivability [SURVmw2]. For functional survival (m), milk yield was included in the model to account for the effect of culling caused by milk production.

Model equation:y = HYS + period + ANIg + PEcow + milk + e/w,(5)
where y—survival record of the period, alternative figure (survival = 100/non-survival = 0);

HYS—herd–year–season fixed effect; 

period—fixed period effect (nine periods); 

ANIg—random additive genetic effect;

PEcow—random effect of the permanent environment with survivability records; 

milk—fixed regression on milk yield variation in the herd for functional survivability (Plemdat, 2020); 

e—random residual effect; 

w—weight, for weighted analysis.

In a statistical model with repeats, all repeats are treated as the same trait, and individuals from different periods are contemporaries of each other in the HYS. Any differences between the periods were handled using a separate effect (period order). The evaluation was performed using the REML method and AIREMLF90 program [35] for unweighted and weighted data. The REML calculation involved a system with approximately 975,000 equations. The REML calculation was terminated when converging to 10^−17^ (convergence criterion). Genetic parameters and effects in the model, including BV, were estimated.

## 3. Results

### 3.1. The Least-Squares Method Fixed Effects

Individual fixed effects explained up to 12% of the variability using the least-squares method (LSM), which reduced the standard deviation of the records in the input dataset from 32.81 to a residual standard deviation of 30.94 (Table 3). HYS and period effects had similar importance at approximately 5%. The HYS + period explained 10% of the variability. Including milk for functional survival increased the explained variability by 12%. The inclusion of sires in the fixed effects did not substantially affect the explained variability, although all effects, including sires, were statistically significant. Statistical significance is related to the dataset size, which was 1.8 million records in our case. We agree with Sewalem et al. [36], who found a significant sire effect, even with a negligible effect on overall variability. In the least-squares method with weights, all fixed effects explained more variability, up to 24%, and the standard deviation decreased from 49.99 to a residual of 43.72. In this case, the sire explained 1% of the variability. The effect on cow survival is strongly conditioned by the breeder’s decision [10], which is included in the HYS effect. The breeder can make systematic decisions [25].

### 3.2. Components of Variance and Genetic Parameters

The genetic parameters are presented in Table 4. The time and number of AIREML iterations required to reach a solution were smaller for models with weights (24 and 25 iterations) than for models without weights (50 and 51 iterations). Convergence criteria, e.g., number of iterations, suggest the stability of the calculation. Lower numbers of iterations mean a higher stability of the calculation.

The mean errors of all parameter estimates were very small, except for the genetic component and the ratios with a genetic component (genetic variance s^2^_G_, heritability h^2^, variance ratio: k = s^2^_r_/s^2^_G_ (residual variance/genetic variance)), where the mean errors were approximately 4% of the estimated value. The phenotypic variance s^2^_P_ in Table 4 corresponded to the standard deviation in Table 3 when fixed effects were considered. In the case without weights, the values of standard deviations were approximately equivalent (without milk: 31.28 compared to 31.26, and with regression to milk: 31.01 compared to 30.99), but not in the case with weights (w2) (55.90 compared to 45.26 and 54.46 compared to 43.88). The inclusion of weights multiplies all variance components. Proportionally, the components for the individual’s permanent environment and the genetic component increased the most. The heritability coefficients for the unweighted calculations were 1.3% and 1.5%, which are consistent with those obtained in studies by van Pelt et al. [13] and Pritchard et al. [21]. For calculations with weights, the heritability was higher—5.8%. Higher heritability of a trait results in a more accurate estimation of breeding value (an additive component of heritability). In the case of weights, an individual’s permanent environment is a significant component, which is essentially zero in cases without weights. Repeatability was 50.5% in the weighted case. An important indicator is the variance ratio k because it indicates the additive component of inheritance. The solution of the system of BLUP equations depends on the ratio of variances. In the statistical models with weights, k was 8.536–9.169, which is several times smaller (more favourable) than in the cases without weights.

### 3.3. Variability of Effects in the Solution

Table 5 shows the summary statistics of the results. These are approximate values where the significance of the effect is inferred from the standard deviations of the individual effects. In the unweighted case, the HYS effect was the strongest, with standard deviations of 10.73 and 10.59 (standard deviations calculated between fixed effect levels). The cow’s survivability is strongly conditioned by the breeder’s decision [10,25]. In the model, the HYS classes could explain this effect. The standard deviations for the period were 8.81 and 8.71 (Table 5).

The values for each period are listed in Table 6. The highest survivability values were for the periods at the beginning of each calving interval and the lowest were at the end calving interval. In absolute terms, the figures for the calculations with weights were higher. The BVs had standard deviations of 1.75 and 2.18 in the case without weights and 6.20 and 6.04 in the case with weights (w2). Figure 2 shows the frequency distribution of breeding values according to the models for survivability [SURV], functional survivability [SURVm], weighted survivability [SURVw2], and functional weighted survivability [SURVmw2]. Although we are evaluating a binary trait, the results of the random effect of BV showed a continuous, near-normal frequency distribution. 

Models with weights were at the peak of the graph (individuals in the pedigree) closer to the mean, and the peaks reached lower values (up to 4%) than in the case without weights (more than 5% and 6%).

The individual’s permanent environment had standard deviations of 33.93 and 32.19 at w2. This shows that when weighting, the permanent environment has the strongest effect. Due to the effect of the weights, there is a substantial increase in the number of observations per individual. The number of records among individuals previously culled later is thus compensated for. The regression coefficient between survivability and milk for functional survivability was 20.13 when calculated without weights and higher at 31.22 when calculated with weights. Despite the fact that the inclusion of milk yield in the model appeared to have only a slight impact on the estimates of the other effects, the actual estimate for this regression is not negligible.

For effects with many levels, correlations among the different calculation methods were calculated for the entire dataset (Table 7). The correlations were generally high, except for the correlation of breeding values between SURVm and SURVw2 (0.75). The high BV correlations among the methods are consistent with the cited data [27].

## 4. Discussion

The presented analysis is limited to the first three parities. The trait under investigation is the cow’s survivability during lactation. In this context, we would like to point out that De Vries and Marcondes [11] stated that the average cow’s productive length of life ranges from 2.5 to 4 years in most developed countries. Despite the recent breeding focus on cow longevity, cows’ actual lifespans are becoming shorter [7]. A functional survivability definition is used in the presented analysis to account for voluntary culling by including milk yield. The distinction between functional and actual survival seems to have become less critical. Maybe the main reason for this phenomenon is that only healthy cows can perform well and reproduce in the Holstein breed. As Schuster et al. [7] stated, more than milk yield, the farmer’s focus is now on the efficiency of the cow. We did not find expressive differences between results for survivability and functional survivability. The increased variability of 12% after including milk is due to increased information in the model. As can be seen in Table 6, there are no distinct differences between the solutions for the survivability and functional analysis. Furthermore, the correlations between the HYS, BV, or PE solutions for analyses with weight w2 are high, above 0.88, as shown in Table 7.

If the survivability assessment is limited to the first three lactations, then there is also an implication for the early prediction of breeding values for the bulls. While the bulls breeding value for milk production is based on the milk production of his daughters in the first three lactations, the estimation of the breeding value for the length of productive life in days is despite the used method delayed because the older the bull’s daughters become, the later the breeding value for longevity is estimated with sufficient accuracy [7]. Therefore, the restriction to the first three lactations allows breeders to simultaneously obtain breeding values for survival during the first three lactations and production traits. In this context, it should be emphasised that the genetic background for survival changes during the lifetime of the animal, as pointed out by van Pelt et al. [13]. The development of the breeding value estimation for longevity has also moved in this direction. Sasaki et al. [37] analysed survival during subsequent lactations with a random regression model in Japanese dairy cattle. Heise et al. [30] followed this up with survival during various periods of the first three parities with multiple-trait linear model analysis. Therefore, using three lactations for the genetic evaluation of survival is in line with the trend of evaluating so-called longevity in dairy cattle.

Genetic parameters of longevity are low; the highest estimates are usually produced by the survival analysis (e.g., [23]; 0.041 on the original scale) and the lowest by the random regression model ([13]; for month intervals 0.002 to 0.031). Although the presented aim of this study was to estimate genetic parameters, the introduction of the weighted analysis of survivability was a vital point of this manuscript. The method used proved to be a suitable procedure for increasing the heritability of the evaluated trait. The values 1.3 % and 1.5 % for the unweighted calculations increased to 5.5% to 5.8% for the weighted analysis. The non-negligible increase of four percentage points was due to the use of weights and the subsequent increase in the number of observations.

The zero variance of the permanent cow environment was due to the nature of the trait being evaluated. A cow culled in the early periods of its life has no further records (it does not repeat any other record). A cow that survived the first period will most likely be culled in the subsequent periods and therefore have the opposite value in the repeated entry. So, there is a low-variance component of an individual’s permanent environment. A zero permanent environment was also reported by van Pelt et al. [13], where the variance of the permanent environment was close to zero for productive lifespan and therefore the effect of the cow’s permanent environment was not included in the evaluation model. On the other hand, the weighted analysis yielded a substantial increase in the variability of the permanent environment, as shown in Table 5. At the same time, the repeatability coefficient and the k-ratio are strengthened.

It is, therefore, clear that the weights used compensated for the information imbalance between animals that were culled at different stages of the first three lactations evaluated. The earlier the animals are culled, the more they are informationally underestimated without using weights. In this information compensation, we see the fundamental advantage of the presented method.

## 5. Conclusions

As already pointed out in the discussion, the main contribution of the weighted method is to compensate for the lack of records in culled individuals. The inclusion of weights partially removed bias from BV survival predictions, as individual differences were more consistent with the expected values. The weighting is multiplied by the variability and coefficient of heritability. As a result of weighting, the most substantial effect became the individual’s permanent environment, and the coefficient of repeatability exceeded 46%. Increasing the coefficient of heritability and taking a significant portion of the residual variability into the individual’s permanent environment changed the ratios between the variance components fitted into the BV prediction equations, thereby creating conditions for a more reliable genetic evaluation. Additionally, the number of REML iterations in the estimation of genetic parameters was lower in models with weights than in models without weights, indicating higher data stability in groups of related animals and groups of fixed effects included in the statistical model.

We used only a simple statistical model with repeatability, which makes it easier to understand why BV predictions are biased. This statistical model has the disadvantage of not fully capturing the genetic basis compared with more complex multi-trait procedures. However, it allows for higher numbers of contemporaries within the HYS, so the period of HYS formation can be shortened and thus better separate the input data from the systematic effect of the farm environment, which tends to have the most substantial effect in statistical models. Future research will focus on validating and assessing the suitability of different statistical models.

## Figures and Tables

**Figure 1 animals-13-01188-f001:**
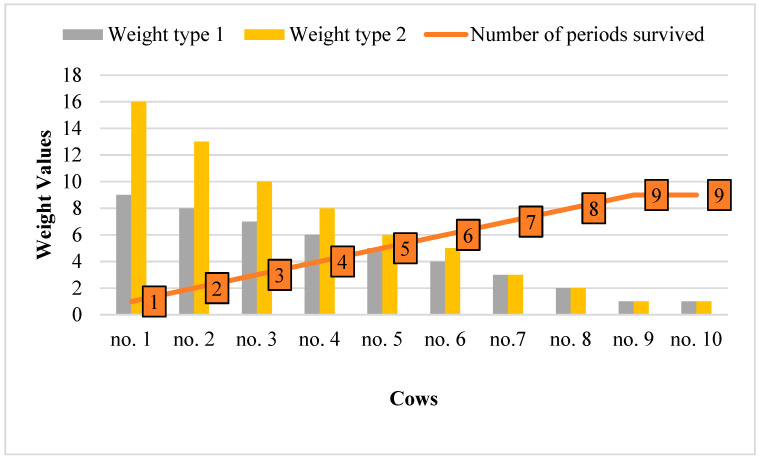
Weight values by the number of periods survived.

**Figure 2 animals-13-01188-f002:**
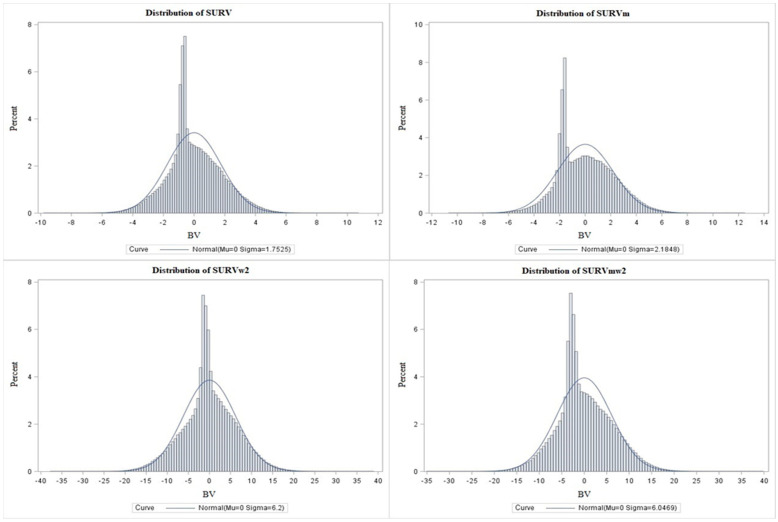
Breeding values frequency distributions: calculations without SURV and SURVm weights and according to level (w2) SURVw2 and SURVmw2 weight.

**Table 1 animals-13-01188-t001:** Weights, breeding value, and reliability of breeding value for 10 different individuals.

Method of Weighting *		Without Weight		w1			w2		
Individual	Failure to Survive a Stage	BV	r^2^	Weight	BV	r^2^	Weight	BV	r^2^
A	1	−0.73	0.02	9	−6.96	0.11	16	−10.09	0.15
B	2	−0.60	0.04	8	−5.42	0.12	13	−7.63	0.15
C	3	−0.47	0.05	7	−3.89	0.12	10	−4.95	0.15
D	4	−0.34	0.07	6	−2.37	0.13	8	−2.76	0.15
E	5	−0.21	0.09	5	−0.85	0.14	6	−0.42	0.15
F	6	−0.09	0.10	4	0.69	0.14	5	1.07	0.15
G	7	0.03	0.12	3	2.25	0.14	3	3.75	0.15
H	8	0.15	0.13	2	3.84	0.15	2	5.34	0.15
I	9	0.26	0.15	1	5.49	0.15	1	6.99	0.15
J	Survival 9	1.99	0.15	1	7.22	0.15	1	8.71	0.15

***** h^2^ = 0.02, survival = 100, non-survival = 0, w1 = weight by number of un-survived periods, w2 = weight by effective number of cases, BV = breeding values, r^2^ = reliability of breeding values.

**Table 2 animals-13-01188-t002:** Description of edited data.

	Statistical Model			
	without Weight		with Weight (w2)	
Cow with records	298,290			
Individuals in the pedigree	660,476			
HYS, records	Number 15,919, Mean size 113.93, Range 15 to 650			
Milk deviation in the herd (SD)	Average 0.99 (0.21), Range 0.06 to 4.60			
Correlation survival rate x milk	0.13		0.22	
**Period**	**Frequency %**	**Discarding % (SD)**	**Weighted Frequency %**	**Weighted Discarding % (SD)**
Average		12.29 (32.8)		49.01 (50.0)
1	16.45	3.64 (18.7)	14.78	37.68 (48.5)
2	15.85	8.69 (28.2)	18.83	55.31 (49.7)
3	14.47	18.67 (39.0)	22.55	69.66 (46.0)
4	11.77	3.43 (18.2)	8.48	22.11 (41.5)
5	11.36	12.74 (33.3)	10.82	46.71 (49.9)
6	9.92	25.61 (43.6)	11.67	63.26 (48.2)
7	7.38	5.35 (22.5)	4.75	14.49 (35.2)
8	6.98	16.56 (37.2)	4.73	28.41 (45.1)
9	5.83	28.60 (45.2)	3.39	28.60 (45.2)
Number of survival records	1,813,636			
Average weight (w2)			1.72	
Sum of weights (w2)			3,119,557	

Frequency % and Weighted frequency % = frequency of surviving cows by period. Discarding % and Weighted discarding % = frequency of discarding cows by period.

**Table 3 animals-13-01188-t003:** Fixed effects—explained variability and survival standard deviations.

Effects Considered	Without Weight		With Weight (w2)	
	SD	R^2^	SD	R^2^
Simple records	32.81		49.99	
Effects of the herd–year–season least-squares method	32.14	0.05	48.13	0.08
Herd–year–season + period	31.26	0.10	45.26	0.18
Herd–year–season + period + milk	30.99	0.12	43.88	0.23
Herd–year–season + period + milk + sire	30.94	0.12	43.72	0.24

SD = standard deviation, R^2^ = coefficient of determination, w2 = weight by effective number of cases.

**Table 4 animals-13-01188-t004:** Population genetic parameters (mean estimation errors in parentheses).

Quantity	Statistical Model			
	SURV	SURVm	SURVSw2	SURVmw2
s^2^_r_	965.97 (1.02)	947.02 (1.00)	1545.40 (1.35)	1529.40 (1.34)
s^2^_G_	12.46 (0.02)	14.60 (0.03)	181.26 (7.56)	166.81 (6.94)
s_PE_	0.00	0.00	1397.70 (6.76)	1269.60 (6.22)
s^2^_P_	978.43 (1.02)	961.62 (1.00)	3124.36 (5.19)	2965.81 (4.78)
h^2^	0.013 (0.0000)	0.015 (0.0000)	0.058 (0.0024)	0.056 (0.0023)
r	0.013 (0.0000)	0.015 (0.0000)	0.505 (0.0009)	0.484 (0.0009)
k = s^2^_r_/s^2^_G_	77.52 (0.1558)	64.54 (0.1302)	8.536 (0.3598)	9.169 (0.3856)

SURV = overall survivability, SURVm = functional survivability, SURVSw2 = overall weighted survivability (weight by effective number of cases), SURVmw2 = functional weighted survivability (weight by effective number of cases), s^2^_r_ = residual variance, s^2^_G_ = genetic variance, s_PE_ = permanent environmental variance, h^2^ = heritability, r = repeatability, k = variance ratio. Mixed model—see Equation (5).

**Table 5 animals-13-01188-t005:** Estimates and solutions of fixed and random effects from mixed-model analyses of unweighted and weighted survival of culling and involuntary culling.

Effects		Statistical Model			
		SURV	SURVm	SURVw2	SURVmw2
Herd–year–season	SD	10.73	10.59	18.37	17.94
	Min	−88.08	−88.92	−105.62	−100.20
	Max	28.02	27.07	57.40	51.34
Periods	SD	8.81	8.71	19.98	19.65
	Min	−13.25	−13.17	−30.48	−30.10
	Max	9.84	9.68	34.69	33.95
Breeding values	SD	1.75	2.18	6.20	6.04
	Min	−10.64	−12.43	−38.79	−39.64
	Max	9.76	10.63	37.46	34.95
Permanent environment	SD	-	-	33.93	32.19
	Min	-	-	−107.96	−137.79
	Max	-	-	85.18	86.68
Milk regression		-	20.13	-	31.22

SD = standard deviation of value, Min = minimum value, Max = maximum value.

**Table 6 animals-13-01188-t006:** REML solution for period survival.

Period	Statistical Model			
	SURV	SURVm	SURVw2	SURVmw2
1	9.43	9.52	34.69	33.95
2	4.48	4.61	19.28	18.86
3	−5.15	−4.92	−6.29	−6.73
4	9.84	9.68	14.80	14.85
5	0.68	0.63	−7.71	−7.49
6	−11.69	−11.45	−30.48	−30.10
7	8.31	7.97	2.63	2.89
8	−2.64	−2.88	−12.33	−12.16
9	−13.25	−13.17	−14.58	−14.06

SURV = overall survivability, SURVm = functional survivability, SURVw2 = overall weighted survivability, SURVmw2 = functional weighted survivability.

**Table 7 animals-13-01188-t007:** Corelation among calculation methods in percent.

	Herd–Year–Season			Breeding Values			Permanent Environment
	SURVm	SURVw2	SURVmw2	SURVm	SURVw2	SURVmw2	SURVmw2
SURV	100	88	89	91	92	94	
SURVm		88	89		75	89	
SURVw2			99			96	99

SURV = overall survivability, SURVm = functional survivability, SURVw2 = overall weighted survivability, SURVmw2 = functional weighted survivability.

## Data Availability

Data are available from the authors.

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
