# Peer review of "Genetic Parameters for a Weighted Analysis of Survivability in Dairy Cattle"

_animals, 2023, doi:10.3390/ani13071188_

Round 1

Reviewer 1 Report

The authors test two different methods of weighting survival data records to compensate for lower reliabilities in animals with early cullings. While this work would be of interest to those in the field of quantitative genetics, some of the results (particularly in Table 2; see comments below) are difficult to interpret. I therefore find it a little difficult to interpret the significance of these results.

Line 103: "important and essential": if something is essential then it's also important by definition

Lines 169 and 172 (Eq 2): wd and w_d would be better written as wd

Line 191: h2 should be written as h2

Line 192: "inverse matrix" of which matrix?

Line 206: r2 should be r2

Table 2: this table is not clear.

* Why do the frequencies (of records/animals?) differ between the weighted and unweighted results?

* Are they not using the same set of records?

* Likewise for the discarding percentage: why does the weighting used affect whether or not an animal/record is discarded?

* Why do (weighted) animals have more records (according to the frequency percentage) for the third period than for the first or second periods, even though a higher percentage are discarded?

* What are the numbers in brackets?

Line 283: I think 10-17 should be written as 10-17

Line 309: "suggests" should be "suggest" ("criteria" is plural)

Line 372: "Models with weights were at the peak of the graph": what does this mean? All four of the graphs shown in Figure 2 have distinct peaks, whether they had weights or not.

Line 446: "used weights" should be "weights used"

Reference 12: the DOI is duplicated

Reference 26: the year is incorrect: the 6th WCGALP conference (Armidale) was in 1998, not 2018 (in Auckland)

Reviewer 2 Report

Animals

Genetic parameters for weighted analysis of survivability in dairy cattle

ÄŒerná, et al.

This is an interesting approach to the difference in information about survival to serial time points to be used as a repeated records approach with regard to survival in dairy cattle.  The approach to weighting the survival (alternatively culling risk) by a function giving more weight to later (full life span) time points makes sense.  Thank you to the authors for providing a fairly well-written and clear manuscript.  However, I have a few points I would like to suggest to make the manuscript clearer to readers that may not be as familiar with survival analyses.

11.        The point of including w1 weights is not clear.  While they are providing weights in the appropriate direction, it is unclear how they can be justified given that only w2 met the goals of weighting the time periods.  I would consider dropping that w1 from the manuscript.  Alternatively, you could leave the w1 in the materials and methods, but state at line 210 “…(w2) yielded better results and only w2 will be considered in the remainder of the manuscript.  So w1 results can be removed from Table2, 3, 4.  And results and discussion. The w1 results are already ignored in most of the discussion.

22.       Please provide for the reader recommendations of how the BV’s should be interpreted.  Should they be back-solved by the weights to provide a meaningful representation of expected progeny differences?  Similarly, the units of measure should be reported for any tables that report REML solutions.

33.       The development of the weights is clear, but there is some confusion in the description.  In Table 1, it appears that the weights are being applied to the records that fail to survive a stage.  But if the records (Line 218) are assigned values of 100 for survival and non-survival values of 0.  Then multiplying the weights by w2 would only weight the survivors records, since w2x0=0.  Then that creates confusion with Table 2.  So, I think it is critical to explain exactly how the weights were applied and to define “Frequency %” and “Discarding %” in Table 2.   The frequency of cows surviving or culled is not changed by whether weighting is used or not.  Representations of the values may change, but then they are no longer the actual frequency % or discarding %.  Maybe it is as simple as changing those titles.

44.       How does this methodology compare to other approaches such as using earlier information to predict survival to specific time point?

Specific edits, comments, and suggestions

Page      Line

1              2              Title.  Suggest “Genetic parameters for a weighted analysis…”  [There can be more than one way, as shown, to define weights].  This is for one set of weights.

1              24           As scientists, I don’t think you “assumed”.  “Given that genetic parameters are expected to change when using weights…”

1              29           Not clear what the two numbers refer to “0.463 or 0.505”.  Need to be specific in abstract.

1              32           “…breeding value’s reliabilities”.  [each is different]

2              58           Only “voluntary culling” is the decision of farmers.

2              60           “production” not “product”  and “…which means an enormous loss of revenue for farmers.”

2              676-69   Please reword.

2              75           “…at 1, 3, 6, and 12 months…”

2              80           “…length of productive life measured in days at 3%, but 5% when measured as the number of lactations initiated.

2              85           Doesn’t the remaining variation also then fall into other correlated traits as well, such as HYS?

2              87-88     I would delete, as this statement belongs in materials and methods or results.

3              116-118                Unclear….reword.

3              136-144                At some point, I think you should indicate why the time periods within lactation are so unequal in length.  It is clearly because of the differences in culling risk in early lactation, but should be pointed out to the reader.

4-152     “…received a weight of two for that period and 1 for periods 1 to 7.”

4              155         Please clarify what is the dependent variable weight is multiplied by.  Some confusion in manuscript about whether it is 100 for survival or 100 for being culled.  An example of a cow including the dependent variable would be very helpful.

4              164         Define how “1” is used in model.  Seems to be a matrix.

4              172         Why “w_d” in model but defined as “wd”?

4              181         Why not define as usual MME with B(beta) for estimates of fixed effects and U for solutions of random effects?

5              195         Table 1 says weights are for periods survived.  Here the second column indicates “Failure to  survive a stage” but indicate the same weights.  Please clarify.

5              195         What are the units for BV?  Is Footnote 1 the assumptions used to derive the values?

5              210         Only report w2 from here on.  If you already determined the results are better, no need to continue reporting on the inferior w1.

5              211 to 219           This seems to fit more in the review of literature in the Introduction section.

5              221         “possible” instead of “passable”

6              224         “…survive all nine periods when data was retrieved in 20XX”. 

6              226-227                How was 15 sires and 15 cows selected?  Given the average was 12 daughters per sire, would 12 or even 10 led to a more representative dataset.  Could any bias have come in by selecting so heavily for the more highly used sires, which may have been represented better genetics and therefor more value?  Did you check for connectedness across sires and HYS?

6              231         The row for HYS is not number.  Just call it HYS.  The size is Mean size.

                                For milk deviation, is the number in  ( ) an SD or SE?

                                Define Frequency % and Discarding %

                              Drop footnote with W1

Title “Description of edited data”

6              248         Probably due to culling open cows.

7              259         Please check style for “Statistical model”

7              261-262                drop w1

8              302         Where did sire come from as fixed effect?  Only HYS and Period and milk were mentioned as fixed effects in the model statement.  What does a fixed sire effect mean when the animal genetic effect is also in the model?

8              311         In Table 4, please indicate the mixed model used in a footnote.

9              349         Table 5 need not mention REML.  That is a method.  Suggest “Estimates and solutions of fixed and random effects from mixed model analyses of unweighted and weighted survival of culling and involuntary culling”.  Please indicate the units the estimates and solutions are in.

10           368         Why the term “Section” as a column heading?  Previously it was called “Period”.  See comment above for more descriptive title.  Indicate units.

11                           In figures , please report units of BV.

11           391         These are Estimates (Fixed) and Solutions (random) from a mixed model using REML.

Round 2

Reviewer 1 Report

The authors' changes have improved the manuscript, and I am now happy for it to be published with some minor proofreading (e.g. on line 71 "epetitive" should be "repetitive").

Reviewer 2 Report

Clarity of manuscript is improved.